# SALVAGE: SHAPLEY-DISTRIBUTION APPROXIMATION LEARNING VIA ATTRIBUTION GUIDED EXPLORATION FOR EXPLAINABLE IMAGE CLASSIFICATION

**Mehdi Naouar**[1,2]**, Hanne Raum**[1]**, Jens Rahnfeld**[1]**, Yannick Vogt**[1,2]**, Joschka Boedecker**[1,2,3]**,**
**Gabriel Kalweit**[1,2]**, Maria Kalweit**[1,2]
[1]University of Freiburg, [2]CRIION, [3]BrainLinks-BrainTools
naouarm@cs.uni-freiburg.de

## ABSTRACT

The integration of deep learning into critical vision application areas has given rise to a necessity for techniques that can explain the rationale behind predictions. In this paper, we address this need by introducing *Salvage*, a novel removal-based explainability method for image classification. Our approach involves training an explainer model that learns the prediction distribution of the classifier on masked images. We first introduce the concept of *Shapley-distributions*, which offers a more accurate approximation of classification probability distributions than existing methods. Furthermore, we address the issue of unbalanced important and unimportant features. In such settings, naive uniform sampling of feature subsets often results in a highly unbalanced ratio of samples with high and low prediction likelihoods, which can hinder effective learning. To mitigate this, we propose an informed sampling strategy that leverages approximated feature importance scores, thereby reducing imbalance and facilitating the estimation of underrepresented features. After incorporating these two principles into our method, we conducted an extensive analysis on the ImageNette, MURA, WBC, and Pet datasets. The results show that Salvage outperforms various baseline explainability methods, including attention-, gradient-, and removal-based approaches, both qualitatively and quantitatively. Furthermore, we demonstrate that our explainer model can serve as a fully explainable classifier without a major decrease in classification performance, paving the way for fully explainable image classification.

## 1 INTRODUCTION

In recent years, the expansion of artificial intelligence (AI) techniques, particularly in the field of computer vision, has revolutionized numerous industries and societal domains, ranging from healthcare to autonomous vehicles. Notably, the emergence of Vision Transformers (ViT), (Dosovitskiy et al., 2021) has lately significantly impacted the field of computer vision, establishing a new standard for image classification. Their ability to leverage self-attention mechanisms and effectively model long-range dependencies has positioned them at the forefront of research and applications across diverse domains. However, despite their remarkable performance, a critical challenge persists: the incompatibility of existing explainability techniques with vision transformers. The majority of explainability methods developed and refined for Convolutional Neural Networks (CNNs) often prove inadequate when applied to vision transformers. This difference highlights the need for architecture-independent explainability methods.

Among such methods, removal-based techniques work by iteratively masking portions of an input image to observe the resulting changes in the model's predictions. If the removal of a particular region significantly affects the model's prediction, it suggests that the region is important for the decision. Conversely, if removing a region has little to no effect, it is deemed less relevant to the model's decision. Recently, ViT Shapley (Covert et al., 2023) has been introduced, merging the principles of removal-based methods with a game-theoretic foundation. The method involves training an explainer model to estimate the Shapley values (Shapley, 1952) of the image patches, quantifying their contribution to the classifier's prediction. The Shapley value of a patch is estimated

as the average change in the model's prediction when the patch is added to the image, calculated across all possible masked variations. Given that the number of possible mask combinations grows exponentially with the number of patches, ViT-Shapley leverages the FastShap (Jethani et al., 2022) algorithm, approximating the Shapley values via a least squares objective and stochastic gradient descent on randomly sampled masks, efficiently handling the computational complexity.

We propose a similar removal-based approach but with a key difference. Instead of approximating the Shapley value during training, we train an explainer model to learn a representation of the classifier's prediction distribution on masked images. At test time, the Shapley value for each patch is then derived from the learned representation. Moreover, we address two weaknesses of ViT-Shapley:

- The ViT-Shapley method often falls short by treating differences in prediction probabilities as linear scores and optimizing them using least squares, which does not adequately capture the probabilistic nature of the classifier's outputs. We address this limitation using a more conventional approach for probability distribution approximations, minimizing the divergence between the explainer's estimated distributions and the classifier's actual prediction distributions.

- The random mask sampling employed by ViT-Shapley is sample-inefficient, especially when dealing with heavy unbalanced ratios of important and unimportant patches, which can hinder effective learning. To mitigate this issue, we adopt an informative sampling strategy to enhance sample efficiency throughout training. By integrating the estimated attribution scores into the sampling process, we are able to achieve a more balanced distribution of masks with low and high prediction likelihoods, thereby facilitating the estimation of underrepresented features.

After incorporating the two proposed optimizations into a method, we refer to as Salvage (Shapley-distribution Approximation Learning Via Attribution Guided Exploration), we performed an evaluation on four datasets (ImageNette, WBC, Pet, MURA) and observed that Salvage outperforms various baseline explainability methods, including attention-, gradient-, and removal-based approaches, both qualitatively and quantitatively. Moreover, we introduce a novel concept, *classifying by explaining* which shifts the focus from explaining a classifier's behavior to aggregating the explainer's estimated feature importance scores into a classification prediction. By doing so we can guarantee the consistency between the predictions and explanations of the model. Our results demonstrate that our explainer can serve as a fully explainable classifier without a major decline in classification performance, advancing the development of more trustworthy image classifiers.

## 2 RELATED WORK

With the increasing demand for explainable AI, a variety of different attribution methods have been explored. These fall into five main categories.

**Class Activation Maps:** Convolutional Neural Networks (CNNs) have inspired the development of Class Activation Mapping (CAM) techniques to highlight important features in visual tasks. The original CAM (Zhou et al., 2015) method works for CNNs with Global Average Pooling (GAP) layers by generating attribution maps based on weighted feature maps. However, this method is limited to architectures with GAP layers. Grad-CAM (Selvaraju et al., 2019) improves upon this by using backpropagated gradients to compute feature map weights, making it more flexible. Variants like Grad-CAM++ (Chattopadhay et al., 2018), Eigen-CAM (Muhammad & Yeasin, 2020), and Ablation-CAM (Desai & Ramaswamy, 2020) explore different ways of refining these weights. Despite these advancements, CAM-based techniques were primarily designed for CNNs and often underperform when applied to transformer architectures (Covert et al., 2023).

**Attention-based Methods:** The attention mechanism of transformer models naturally allows insights into the information flow within the network. A straightforward method to assess importance is by analyzing the attention scores between the class token and input tokens at a specific layer (Clark et al., 2019). However, this approach gives limited insight into the overall information flow since different layers may focus on different regions, and the final output is shaped by the interaction across all layers. (Abnar & Zuidema, 2020) tackles this by modeling the information flow as a directed acyclic graph, using attention scores as edge weights. They propose two methods to extract input token relevance: attention rollout, which traces attention weights from the class token back to the input tokens, and attention flow, which estimates information flow using maximum flow computations in the graph. However, attention mechanisms often exhibit issues like high attention

scores focusing on low-informative background regions (Covert et al., 2023; Darcet et al., 2023). (Darcet et al., 2023) suggest this problem stems from the use of random tokens as intermediaries for internal computations and address it by adding supplementary tokens. While attention scores can be useful in some cases, recent studies question their reliability as explanations, arguing they may not always reflect a model's true reliance on each token (Jain & Wallace, 2019; Serrano & Smith, 2019). Moreover, attention-based methods are class-agnostic, providing a single explanation per prediction and lacking class-specific insights.

**Gradient-based Methods:** Saliency maps (Simonyan et al., 2014) are an early method that extracts the gradient of the class score with respect to the input image. However, these gradients can be highly sensitive to small input perturbations, resulting in significant fluctuations (Smilkov et al., 2017). To mitigate this, SmoothGrad (Smilkov et al., 2017) averages the gradients over multiple noisy versions of the input image, effectively smoothing the gradients and reducing volatility. Integrated Gradients (Sundararajan et al., 2017) further improves on this by integrating gradients along the path between a baseline image (typically a black image) and the target image.

**Removal-based Methods:** Treating neural networks as a complete black box function, removal-based Methods measure fluctuations in the predicted class probabilities under partial information. The estimation of the prediction under partial information is achieved by inferring the classifier on masked images. RISE (Petsiuk et al., 2018) suggests measuring the contribution of each part of the image by sampling a large number of masks, computing the prediction of the network on each masked image, and finally summing up the averaged product of the masks with their corresponding predictions. However, as the number of possible masks grows exponentially in the number of image patches, a large amount of masks is required to obtain a decent estimation for each single region. To address this issue, FastSHAP (Jethani et al., 2022) proposes a game theory approach, training an explainer model to estimate the Shapley value of the image patches, which consists of the average change in the model's prediction when the patch is added to the image. Building upon this concept, ViT-Shapley (Covert et al., 2023) further extends this method by adopting a vision transformer-based architecture for the explainer model. While this method aims to train a model to approximate the Shapley value directly, our approach learns a representation of the classifier's prediction distribution from which the Shapley value can be extracted.

**LRP-based Methods:** Layer-Wise Relevance Propagation (LRP) (Lapuschkin et al., 2015), based on Deep Taylor Decomposition (DTD) (Montavon et al., 2017), explains model predictions by propagating the output back to the input using specific decomposition rules. While LRP has shown good results on CNNs, applying it to transformer architectures has led to unstable explanations. (Chefer et al., 2020) attributes these instabilities to skip connections and attention layers, and the authors propose alternative propagation rules for these operations, particularly combining LRP relevance with gradient-based attention rollout for attention layers. Similarly, (Ali et al., 2022) attempts to address this issue by proposing more stable rules for the self-attention and LayerNorm operations.

## 3 BACKGROUND

In this section, we introduce Shapley values (Shapley, 1952), which serve as the foundation of our method. Originating from cooperative game theory, Shapley values are used to fairly distribute payoffs among players based on their individual contributions to the total value. In the context of machine learning, they quantify the impact of each feature on a model's prediction by measuring the average change in prediction when the feature is included in an input subset. We begin by presenting the formal definition of Shapley values, followed by a rearrangement that enables their approximation without requiring their marginal contributions.

### 3.1 SHAPLEY VALUES

Let $N$ be a set of features and $v(S)$ the prediction outcome given a feature subset $S \subset N$. The Shapley value $\phi_i$ of a feature $i$ is obtained as follows:

$$\phi_i = \sum_{S \subseteq N \setminus \{i\}} \underbrace{\frac{|S|!\,(|N| - |S| - 1)!}{|N|!}}_{w_S} \left( v(S \cup \{i\}) - v(S) \right) \tag{1}$$

The computation of Shapley values for all features requires evaluating the predictions for every possible subset of features. However, as the number of features increases, the number of subsets grows exponentially, making the computation of the exact Shapley values infeasible for very large numbers of features. To address this challenge, FastShap (Jethani et al., 2022) proposes an approximation of the Shapley values using a least-squares objective over the feature subset distribution $p_w(S) \propto w_S$, sampled proportionally to $w_S$ :

$$\mathbb{E}_{p_w(S)}[(v(S) - \sum_{i \in S} \phi_i)^2] \tag{2}$$

However, using the Mean Squared Error (MSE) loss to approximate probabilistic model outputs introduces significant limitations. MSE is primarily designed for comparing scalar values and is not well-suited for capturing the complexities of probability distributions, such as their inherent uncertainty, variance, and multimodal characteristics. Additionally, MSE is sensitive to scale and does not enforce the necessary constraints of probability measures, like non-negativity and normalization. As a result, MSE often yields invalid or suboptimal approximations when applied to distributions.

## 3.2 Approximating Shapley Values without Marginal Contributions

Kolpaczki et al. (2024) suggests a rearrangement of the Shapley value formula. Instead of expressing it as the weighted average of marginal contributions, it can be viewed as the difference between the weighted average of prediction outcome when feature $i$ is included and the weighted average of prediction outcome when feature $i$ is excluded:

$$\phi_i = \underbrace{\sum_{S \subseteq N \setminus \{i\}} w_S \cdot v(S \cup \{i\})}_{\phi_i^+} - \underbrace{\sum_{S \subseteq N \setminus \{i\}} w_S \cdot v(S)}_{\phi_i^-} \tag{3}$$

The positive and the negative Shapley values can be seen as the expected values $\phi_i^+ = \mathbb{E}[v(S \cup i)]$ and $\phi_i^- = \mathbb{E}[v(S)]$, over the set distribution $p_w(S) \propto w$ for $S \subseteq N \setminus \{i\}$.

## 4 Approach

### 4.1 Shapley Distribution Estimation

We build upon the concept of optimizing Shapley values without relying on marginal contributions by training an explainer model to learn both positive and negative Shapley values. During training, we sample masked images from the distribution $p_w(S) \propto w_S$. For each sampled masked image, we update the estimated positive Shapley values $\phi_i^+$ for all visible image patches $i \in S$, and the negative Shapley values $\phi_j^-$ for all masked patches $j \notin S$. This is achieved by minimizing the difference between the sum $\sum_{i \in S} \phi_i^+ + \sum_{i \notin S} \phi_i^-$ and the actual prediction outcome $v(s)$. As mentioned in section 3.1, using the mean squared error (MSE) to approximate the target distribution $v(s)$ would yield suboptimal results because of the probabilistic nature of the classifier's output. Therefore, we propose mapping the summed term into a probability distribution $u(S)$, which we refer to as *Shapley probability distribution*:

$$u(S) = \sigma(\sum_{i \in S} \phi_i^+ + \sum_{i \notin S} \phi_i^-) \tag{4}$$

where $\sigma$ denotes the softmax function in a multiclass classification setting or the sigmoid function for binary classification. The Shapley distribution of the masked image is then optimized by minimizing the Jensen–Shannon (JS) (Lin, 1991) divergence between the classifier's prediction $v(S)$ and its corresponding estimated probability distribution $u(S)$:

$$\underset{\phi^+, \phi^-}{\arg\min} \; \mathbb{E}_{p_w(S)}[D_{JS}(u(S)||v(S))] \tag{5}$$

At test time, the feature importance scores of each feature (image patch) are given by their estimated Shapley value $\phi_i = \phi_i^+ - \phi_i^-$.

## 4.2 Attribution Guided Sampling

In our experiments, we observed that sampling from the random mask distribution $p_w$ often results in a disproportionate number of masked images having either high or low likelihoods of the predicted class. This imbalance severely affects the minority class estimation, since a large number of samples are required to estimate the values of its members. This finding motivated us to address this imbalance through an alternative mask distribution, thereby enhancing the sampling efficiency during training. Thus, we propose exploiting the current estimates of feature importance scores $\phi$ to rebalance the ratio of masks, targeting the most and least informative regions of the image. Our proposed informative sampling distribution, $p_\phi(S) \propto \phi$, operates in two stages:

1. Sampling the number of masked patches: First, the number of masked patches within an image is sampled from a uniform distribution $U(1, n)$, where $n$ represents the total number of patches in the image.
2. Selecting patches: Next, patches are selected without replacement, using the estimated feature importance scores $\phi$ of the target class, as sampling weights. For instance, if three patches have importance scores of 0, 1, and 3, their corresponding probabilities of being sampled would be 0, 0.25, and 0.75, respectively.

We then generate two equal mask subsets: the first, prioritizing the most informative regions from the image, and the second by masking them to target the least informative regions. A detailed description of the sampling process is provided as pseudo-code in Algorithm 3. Compared to random sampling, $p_\phi(S)$ yields a mask distribution with more balanced prediction likelihoods (see Figure 6), thereby enhancing sample efficiency during training.

## 4.3 From an Explainer to a fully explainable Classifier

Since the classifier and the explainer are two decoupled models, the explainer merely approximates the behavior of the classifier. Thus, there is no guarantee of consistency between the classifier's predictions and the explainer's explanations, especially under domain shift. We suggest addressing this issue by using the explainer as a unified model for both classification and explanation.

Recall that Salvage is trained to minimize the divergence between its Shapley distribution $u(S)$ and the corresponding classifier prediction $v(S)$. By setting $S$ to the full (unmasked) image $N$ in eq. (4), we obtain the explainer's approximation for the classifier's prediction for the complete image:

$$u(N) = \sigma(\sum_{i \in N} \phi_i^+) \approx v(N) \qquad (6)$$

In addition to ensuring consistency between the classification prediction and its explanation, using the explainer as a classifier offers a unique advantage. By aggregating the importance scores of each image region, we obtain a precise understanding of how each region contributes to the overall prediction. This approach results in a classifier that is fully transparent and explainable.

## 5 Experiments

In this section, we first describe our experimental setup. We then conduct both a qualitative and a quantitative analysis of our method and several baselines. Next, we conduct an ablation study, showing the advantage of informative sampling. Finally, we evaluate the classification and explanation of Salvage as an explainable classification method.

## 5.1 Experimental Setup

We adopt the experimental setting from ViT-Shapley (Covert et al., 2023), evaluating the explanation performance of our method across the ImageNette (Howard & Gugger, 2020) , WBC (bodzás et al., 2023), and Pet (Parkhi et al., 2012) datasets for multi-class classification and the MURA (Rajpurkar et al., 2018) dataset for binary classification. For the target model, we use a Vision Transformer base model (ViT-B) with a patch size of 16 and registers (Darcet et al., 2023), leveraging the implementation and pre-trained weights provided by DINOv2 (Oquab et al., 2024). The

classifier model is trained for 25 epochs and fine-tuned on masked images for 50 epochs. We adopted the Segmenter (Strudel et al., 2021) segmentation architecture for our explainer models and trained them using 32 masks per image for $\sim 18k$ iterations (corresponding to 50 epochs for ImageNette, WBC, and MURA, and 100 epochs for Pet). All models were trained with a batch size of 64, an AdamW (Loshchilov & Hutter, 2019) optimizer with a learning rate of 1e-5 and a weight decay of 1e-5, except the explainer models of ViT-Shapley which were trained with lr 1e-4. We compare our method to 10 different baselines; GradCam (Selvaraju et al., 2019), EigenCam (Muhammad & Yeasin, 2020), Attention scores from the last layer with registers (Clark et al., 2019; Darcet et al., 2023) (Attn. last), Attention Rollout with registers (Abnar & Zuidema, 2020; Darcet et al., 2023), ViT-CX (Xie et al., 2023), Saliency maps (Simonyan et al., 2014), Integrated gradients (Sundararajan et al., 2017), LRP beyond attention (Chefer et al., 2020), RISE (Petsiuk et al., 2018), and ViT-Shapley (Covert et al., 2023). As a reference, we additionally evaluate the metric scores on randomly generated maps (Random). The implementation of the CAM-based methods is adapted from (Gildenblat & contributors, 2021), while the other baselines use their original implementations.

## 5.2 QUALITATIVE ANALYSIS

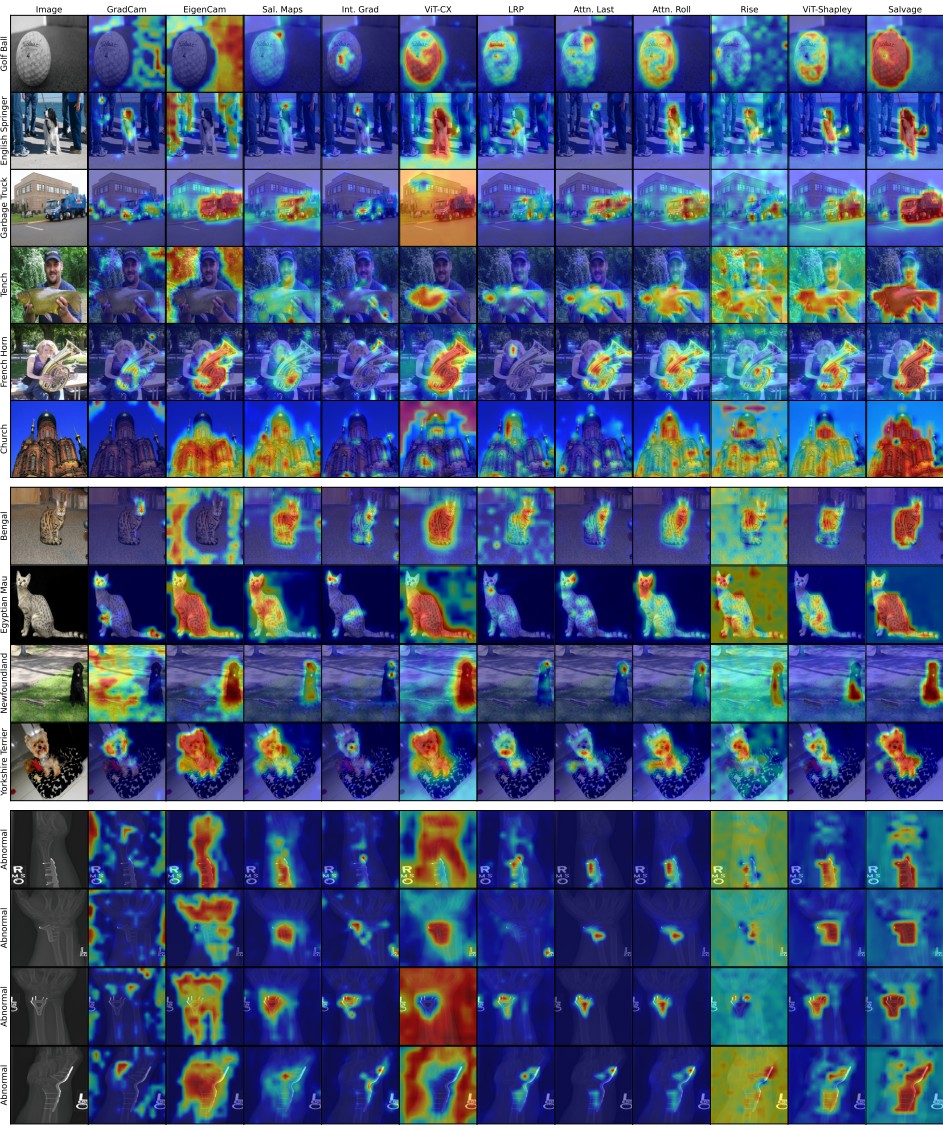

Figure 1: Qualitative examples computed on ImageNette, Pet and MURA.

We conduct a qualitative analysis comparing our method with the baselines. We present 6 examples from ImageNette, 4 from Pet, and 4 from MURA in Figure 1. Saliency maps, Vit-CX, and LRP seem unreliable as their attribution maps often focus on random parts of the background. Despite its solid quantitative results, the attribution maps generated by RISE are highly noisy. Attention rollout (with registers) and ViT-Shapley yield decent results on most images with relatively low scores outside the informative regions. Solely Salvage excels at highlighting the entire relevant region of interest. Further qualitative examples from ImageNette, Pet and MURA, as well as qualitative examples from WBC, can be found in Appendix A.5.

## 5.3 QUANTITATIVE ANALYSIS

Table 1: Quantitative results computed on the Pet, ImageNette, WBC, and MURA datasets. The performance of the 10 compared methods is measured in terms of SRG, R-SRG, RMA, and RRA.

| Method | Pet | | | | ImageNette | | MURA | | WBC | |
| | SRG | R-SRG | RMA | RRA | SRG | R-SRG | SRG | R-SRG | SRG | R-SRG |
|---|---|---|---|---|---|---|---|---|---|---|
| GradCam | 10.6 | 3.5 | 48.1 | 42.7 | -1.9 | -3.3 | 16.2 | 10.1 | -18.5 | -20.2 |
| EigenCam | 27.4 | 3.2 | 48.9 | 62.9 | 13.2 | -3.1 | 0.1 | -4.5 | 22.9 | -7.0 |
| Attn. last | 47.9 | 9.6 | 61.1 | 70.1 | 27.0 | 3.0 | 22.4 | 7.0 | 42.2 | 1.6 |
| Attn. Roll. | 52.0 | 11.2 | 51.5 | 74.6 | 32.0 | 3.4 | 17.6 | 6.3 | 48.1 | 2.5 |
| ViT-CX | 50.2 | 17.6 | 30.6 | 67.5 | 29.9 | 7.5 | 19.8 | 9.1 | 41.6 | 7.3 |
| Sal. Maps | 51.1 | 10.8 | 52.7 | **76.3** | 27.7 | 2.8 | 25.3 | 8.5 | 42.8 | 2.1 |
| IntGrad | 27.4 | 7.9 | 51.5 | 58.8 | 11.0 | 2.2 | 13.9 | 6.1 | 11.8 | 1.6 |
| LRP | 49.5 | 9.2 | 63.9 | 71.8 | 27.9 | 3.0 | 19.3 | 6.8 | 37.0 | 1.7 |
| RISE | 63.7 | 18.5 | 30.1 | 47.8 | 22.9 | 5.4 | 56.5 | 22.1 | 20.7 | 3.5 |
| ViT-Shap | 61.1 | 14.7 | 52.7 | 69.0 | 40.3 | 6.2 | 65.3 | 20.6 | 57.4 | 7.3 |
| Salvage | **68.5** | **26.3** | **64.9** | 73.5 | **51.3** | **14.9** | **68.6** | **25.3** | **69.7** | **22.6** |
| Random | 0.0 | 0.0 | 30.0 | 29.4 | 0.0 | 0.0 | 0.0 | 0.0 | 0.0 | 0.0 |

Based on our baselines, we employed various metrics to assess Salvage's performance. Since the true importance of features is unknown beforehand, evaluating explanation accuracy poses a challenge. The metrics used to evaluate our method include Most and Least Influential First (MIF, LIF) (Petsiuk et al., 2018), Symmetric Relevance Gain (SRG) (Blücher et al., 2024), Relevance Rank Accuracy (RRA) and Relevance Mass Accuracy (RMA) (Arras et al., 2022). MIF and LIF (also known as Deletion and Insertion) measure performance by progressively removing image patches based on their importance scores, with the goal of minimizing MIF (removing most important features first) and maximizing LIF (removing least important features first). SRG improves upon these metrics by addressing their sensitivity to masking strategies and calculating the difference between the MIF and LIF scores to provide more consistent performance rankings. Additionally, RRA and RMA assess the alignment of feature importance scores with human-annotated regions of interest. RRA evaluates how well the top-k important patches match the annotated region, while RMA measures the proportion of attributions within the annotated area, reflecting the focus on relevant regions.

It is important to note that MIF, LIF, and SRG offer only limited insights into the quality of explanations, as they focus solely on the ranking of features while disregarding their relative differences. To address this limitation, we extend our metric selection by further including three new metrics to assess the relative score differences within an attribution map. Specifically, we extend the MIF and LIF metrics into R-MIF (Relatively Most Influential First) and R-LIF (Relatively Less Influential First). Instead of selecting patches purely based on their rank, we use estimated importance scores as weights in a sampling process to determine which patches to add first. To reduce sampling variance, we generate 128 masks for each size of feature subset. Analogously to SRG, we define R-SRG as the difference between R-LIF and R-MIF. These proposed metrics provide deeper insight into the relative importance of feature attribution scores and their influence on the model predictions. A pseudo-code and a more detailed description of the metrics is provided in appendix A.1.

Our results, as shown in Table 1 and Figure 2, clearly demonstrate that our method outperforms all baselines across both rank-based and relative-based metrics on all three datasets. We have observed in Figure 2 that all top-3 methods (Rise, ViT-Shapley, and Salvage) reach similar LIF and R-LIF scores – hinting, that all three methods highlight a small subset of important features which is sufficient for the model to recognize the classified object. However, we observed that Salvage reaches significantly better MIF and R-MIF scores, which suggests that our method is more effective at identifying a larger portion of the important features compared to the other methods. Disregarding the top-3 methods, our results are in line with Covert et al. (2023) showing poor performance of the CAM-based, attention-based, gradient-based, and LRP-based methods. The LIF, MIF, R-LIF, and R-MIF scores of all methods have been included in appendix A.3 for the sake of completeness.

Moreover, we evaluate the RMA and RRA metrics on the Pet dataset, for which human-annotated regions of interest were provided. The results (cf. Table 1) showed that Salvage achieves the best scores in terms of RMA. We additionally observe that Sal. Maps and Attn. Roll. reach slightly higher RRA scores showing a strong alignment in ranking, meaning their top attribution scores lie within the object of interest. However, these methods also show a lower attribution mass within the object (RMA) and exhibit weaker SRG and R-SRG scores, suggesting they may not focus on the most relevant parts of the object for the prediction, potentially limiting their effectiveness.

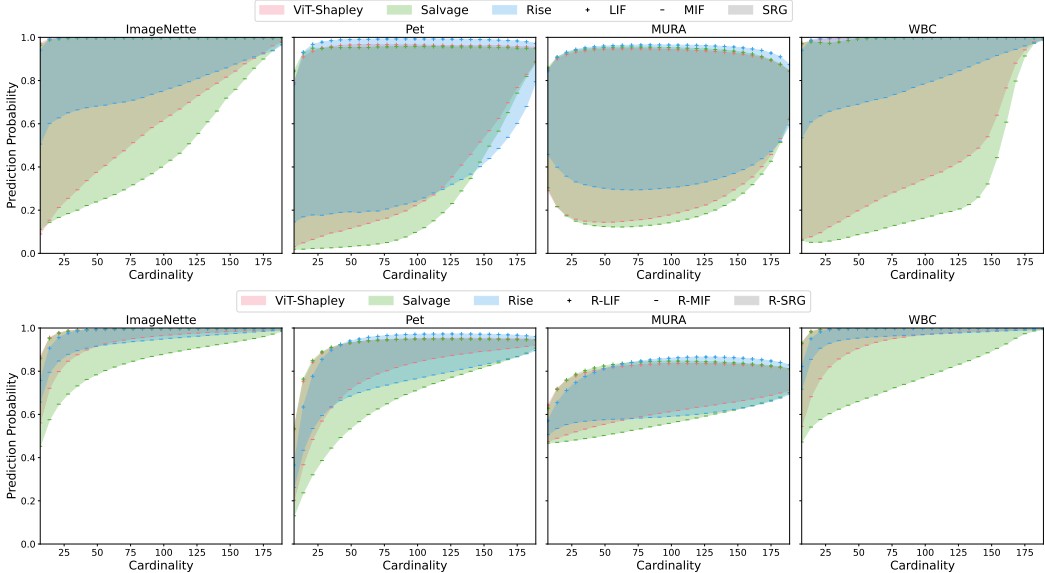

Figure 2: An Illustration of the MIF, LIF, SRG, R-MIF, R-LIF, and R-SRG across the different cardinality values (number of masked patches) for the top-3 performing methods (RISE, ViT-Shapley, and Salvage) computed on the Pet, ImageNette, WBC, and MURA datasets.

## 5.4 ABLATION STUDIES

In this section, we investigate the individual contribution of each core principle in our method, Shapley distribution estimation and informative sampling, through two ablation studies. The first study compares the performance of Shapley distribution estimation to the MSE-based approximation (ViT-Shapley) without informative sampling. The second study investigates the effect of informative sampling on the performance of Salvage by comparing its performance with and without the use of informative sampling.

The results of our studies, illustrated in Figure 3, reveal the following: (a) Shapley distribution estimation outperforms MSE-based approximation (ViT-Shapley) on ImageNette, WBC, and Pet datasets, but shows a slightly lower performance on MURA. However, we believe the latter could benefit from adjusting the temperature parameter in the sigmoid function used during training. (b) Informative sampling clearly improves performance on MURA and ImageNette, while achieving modest gains in R-SRG of 0.68 and 0.26 for WBC and Pet, respectively.

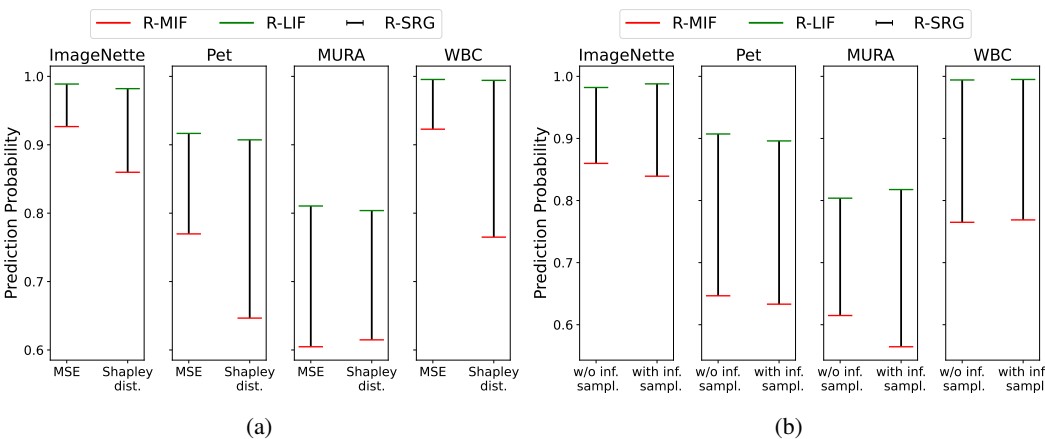

(a)                                                    (b)

Figure 3: (a) Ablation study quantifying the performance gain of using Shapley distribution esti-mation versus MSE-based approximation (ViT-Shapley). (b) Ablation study comparing the perfor-mance of Salvage with and without informative sampling. The performance is reported in terms of R-SRG, which is given by the difference between R-LIF and R-MIF (the larger the better).

## 5.5 FROM EXPLAINER TO EXPLAINABLE CLASSIFIER

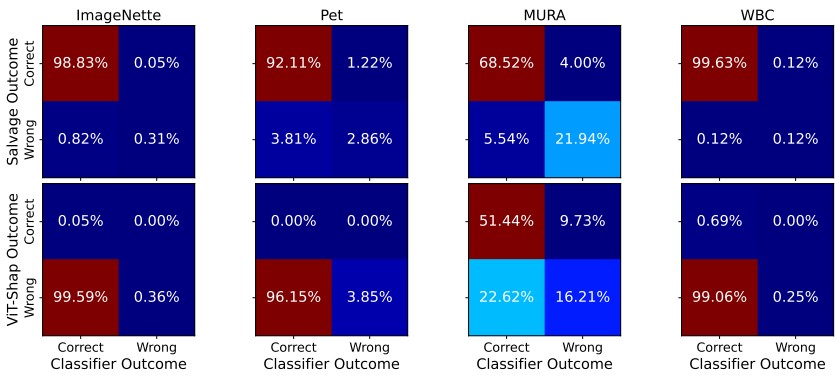

Figure 4: Confusion matrices showing the overlap between correctly and incorrectly classified sam-ples for the explainer models (Salvage and ViT-Shapley) and classifier. Each cell indicates the percentage of samples classified correctly or incorrectly by both models, with rows representing the explainer's outcomes and columns representing the classifier's outcomes.

In this subsection, we assess the benefits of using Salvage as an explainable classifier. By directly deriving predictions from the feature importance scores, the exact contribution of each image re-gion to the classification outcome is made explicit. This makes Salvage particularly well-suited for applications where explainability is as critical as classification accuracy.

We start by analyzing the overlap between correctly and incorrectly classified samples for the ex-plainer and classifier in Figure 4. The results suggest that Salvage demonstrates no major drop in performance relative to the original classifier and a high agreement with the predictions of the classifier. In contrast to Salvage, we observe that the ViT-Shapley explainer yields a poor classifica-tion performance and low overlap with the predictions of the classifier. A more detailed qualitative analysis of the classification performance of Salvage is presented in Table 4.

Next, we analyze the explanations of Salvage in cases where its classification prediction is inaccu-rate. In Figure 5, we present examples where Salvage failed to produce the correct classification prediction and provide its attribution maps for both the predicted class (second row) and ground truth classes (third row). In the second row of the figure, we can see that Salvage provides clear, understandable explanations for its misclassifications. For example, it can explain errors such as

mistaking a monument for a church, a construction truck for a garbage truck, or a mouse for an English Springer Spaniel. Notably, in the last row, even in these failure cases, the attribution map corresponding to the ground truth class of the image still accurately highlights the ground truth object (parachute, french horn,...). The ability to provide reliable explanations, even in cases of classification failure, underscores Salvage's strong potential to serve as a fully explainable classifier.

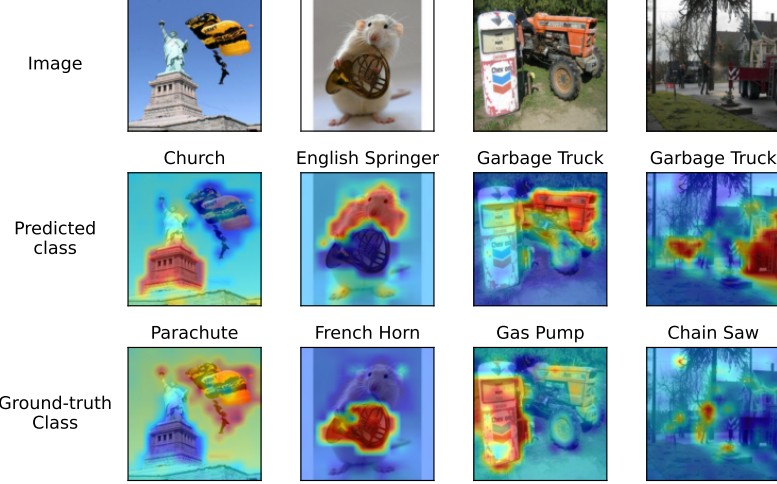

Figure 5: Examples of test images from ImageNette where Salvage fails to make a correct classification prediction. In the second row, we present the Salvage's attribution map for the (wrongly) predicted class, in the last row its attribution map for the ground truth class of the image.

## 6   CONCLUSION

By incorporating a novel methodology for attribution score estimation and informative sampling, we have developed a removal-based explanation method for image classification called Salvage. Our experiments demonstrate that Salvage outperforms all 10 evaluated baselines, both qualitatively and quantitatively, by delivering higher-quality explanations and clearly distinguishing relevant from irrelevant image regions. Beyond its strong explanation performance, we have also established Salvage's potential as a fully explainable classifier. While its classification accuracy is comparable to that of a classifier model, Salvage consistently provides detailed and interpretable explanations, even for images that are misclassified. This capability not only highlights the regions contributing to the predictions but also helps users understand the underlying factors leading to errors. Overall, these features underscore Salvage's strong potential to serve as a fully explainable classifier in applications where explainability is as critical as classification accuracy.

## 7   LIMITATIONS AND FUTURE WORK

Our work offers several promising avenues for further advancement. Future optimizations of our method could involve refining the neural architecture of the explainer model, as improved segmentation architectures may enhance its ability to capture spatial relationships and accurately estimate attribution scores for each superpixel-class pair. Moreover, introducing a temperature parameter in the softmax or sigmoid functions during the approximation of the classifier's distribution may be valuable and could offer a better alignment of the magnitude of the approximated values with the output logits of the classifier. Additionally, since Salvage adopts the principles of Shapley's additive explanation, it relies on the assumption that all features are linearly independent—an assumption that may be overly restrictive in practice. A promising direction for future work could involve extending Salvage to account for feature interactions. Motivated by the quality of the explanation maps produced by Salvage, we plan to explore its potential on different tasks and data modalities, as well as an unsupervised segmentation model in future research.

ACKNOWLEDGMENTS

This project was funded by the Mertelsmann Foundation and the Deutsche Forschungsgemeinschaft (DFG, German Research Foundation) – Project-ID 499552394 – SFB 1597. This work is part of BrainLinks-BrainTools which is funded by the Federal Ministry of Economics, Science and Arts of Baden-Württemberg within the sustainability program for projects of the excellence initiative II.

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

# A    APPENDIX

## A.1    METRICS OVERVIEW

**Least influential First (LIF) and Most influential First (MIF)** : also known as Insertion and Deletion (Petsiuk et al., 2018) repeatedly mask the images by removing image patches based on their ascending/descending ranking from the least important to the most important attribution scores of the method being evaluated. The area under the resulting curve (predictions/number of patches) is then computed as the performance score as illustrated in Algorithm 1. The LIF score should be maximized, which is achieved by removing the least important features first, in order to get high predictions with as few patches as possible. Conversely, the MIF score should be minimized by removing the most influential patches first. It is important to note that MIF and LIF offer only limited insights into the quality of explanations, as they focus solely on the ranking of features while disregarding their relative differences.

---

**Algorithm 1** Most Informative First (MIF) and Least Informative First (LIF)

---

**Require:** $N$: Set of all features, $v$: Prediction outcome function, $\phi$: Importance scores for each feature, $metric$: 'MIF' or 'LIF'
**Ensure:** (averaged) MIF or LIF score
1: $S \leftarrow \emptyset$                                               ▷ Initialize empty subset
2: $preds \leftarrow []$                                            ▷ List to store prediction outcomes
3: **if** $metric$ = 'LIF' **then**
4:     $sorted\_features \leftarrow$ Sort features in descending order of importance based on $\phi$
5: **else**
6:     $sorted\_features \leftarrow$ Sort features in ascending order of importance based on $\phi$
7: **end if**
8: **for** $i = 1$ to $|N|$ **do**
9:     Add $sorted\_features[i]$ to $S$
10:     $pred \leftarrow v(S)$                   ▷ Evaluate the prediction outcome with the current subset
11:     Append $pred$ to $preds$
12: **end for**
13: $final\_score \leftarrow \frac{\sum \text{preds}}{|N|}$                        ▷ Compute the average over all subset sizes
14: **return** $final\_score$

---

**Symmetric Relevance Gain (SRG)**: In a recent study, Blücher et al. (2024) demonstrated the inconsistency of the MIF and LIF metrics while using different masking strategies, as these can lead to different performance rankings depending on the robustness of the masking strategy. In order to address this issue, the authors presented a simple, yet effective metric named SRG, which is given by the difference between the MIF and LIF scores:

$$SRG = LIF - MIF \qquad (7)$$

The authors have shown on 40 different masking strategies that this metric breaks the inherent connection to the underlying occlusion strategy and leads to consistent rankings.

**Relatively Most influential First (R-SRG), Relatively least influential First (R-LIF)** MIF, LIF, and SRG provide limited insights into the quality of explanations, as they focus exclusively on feature ranking without accounting for the relative difference scores between the features. To address

this limitation, we propose an extension of these metrics aimed at capturing the faithfulness of the relative differences in importance scores across different features. While MIF and LIF evaluate the prediction model by selecting the patches to be removed purely based on their ranking, we propose selecting $n$ features through a weighted sampling process. This process uses feature importance scores as sampling weights, ensuring that features are sampled in proportion to their estimated importance. By employing this method, we can assess the faithfulness of the relative differences in attribution scores, which are integrated into the sampling process. Analogous to MIF and LIF, R-MIF aims to sample the most relevant features, while R-LIF aims to sample the least relevant ones. In our experiments, we repeated the sampling process for each mask size 128 times to minimize variance in performance scores resulting from the sampling process ($n_{masks} = 128$). For a more detailed overview, we present the pseudo-code for the computation of the R-MIF and R-LIF scores in Algorithm 2.

---

**Algorithm 2** R-MIF and R-LIF

---

**Require:** $N$: Set of all features, $v$: Prediction outcome function, $\phi$: Importance scores for each feature, $metric$: 'R-MIF' or 'R-LIF', $n\_masks$: Number of subsets to sample per cardinality
**Ensure:** (averaged) R-MIF or R-LIF score
  1:   $S \leftarrow \emptyset$                                                         ▷ Initialize empty subset
  2:   $preds \leftarrow []$                                        ▷ List to store averaged prediction outcomes
  3:   $\phi\_min \leftarrow \min(\phi)$                                ▷ Find minimum feature importance score
  4:   $\phi\_max \leftarrow \max(\phi)$                              ▷ Find maximum feature importance score
  5:   $\phi\_norm \leftarrow \frac{\phi - \phi\_min}{\phi\_max - \phi\_min}$       ▷ Min-max normalization ensuring positive sampling weights
  6: **if** $metric$ = 'R-LIF' **then**
  7:     $weights \leftarrow \phi\_norm$                 ▷ Use the normalized scores as weights for R-LIF
  8: **else**
  9:     $weights \leftarrow 1 - \phi\_norm$        ▷ use 1 minus the normalized scores as weights for R-MIF
10: **end if**
11: **for** $i = 1$ to $|N|$ **do**
12:     $subset\_preds \leftarrow []$               ▷ Store prediction outcomes for this step
13:     **for** $j = 1$ to $n\_masks$ **do**
14:        $S_j \leftarrow$ draw $i$ features from $N$ using $weights$ as sampling weights, without replacement
15:        $pred \leftarrow v(S_j)$        ▷ Evaluate the prediction outcome with the sampled subset
16:        Append $pred$ to $subset\_preds$
17:     **end for**
18:     $avg\_pred \leftarrow \frac{\sum subset\_preds}{n\_masks}$       ▷ Average the prediction outcomes over all samples
19:     Append $avg\_pred$ to $preds$
20: **end for**
21: $final\_score \leftarrow \frac{\sum preds}{|N|}$       ▷ Compute the final average score over all subset sizes
22: **return** $final\_score$

---

**Relative Symmetric Relevance Gain (R-SRG)** Analogously to SRG, we combine the R-MIF and R-LIF scores by defining:

$$R\text{-}SRG = R\text{-}LIF - R\text{-}MIF \tag{8}$$

**Relevance Rank Accuracy (RRA) / Relevance Mass Accuracy (RMA)** Arras et al. (2022) presented two metrics measuring the consistency of the feature importance scores with a target region of interests provided by human annotations. For relevance rank accuracy, image patches are ordered based on their importance scores, and the number of top-k pixels within the ground truth mask is measured, with k set to be the number of pixels in the ground truth mask. A high relevance rank score is indicative of a strong alignment between the explanation and the human annotation. For relevance mass accuracy, the ratio of positive attributions within the ground truth mask to the sum of all positive attributions is calculated. A high relevance mass score indicates that significant attention is placed on the same region as the human annotation, with little attention directed to other regions.

### A.2 SAMPLING DISTRIBUTION

For the sake of completeness, we present a pseudo code of the informative sampling procedure in Algorithm 3. Moreover, we computed the average prediction of the sampled subsets once using ran-

dom sampling and once using attribution-informed sampling. The results illustrated in appendix A.2 demonstrate that the average prediction likelihood of our informative sampling technique yields a more balanced distribution than random sampling.

Additionally, we evaluated the effect of addressing this unbalance on sample efficiency. To do so, we illustrate in Figure 7 the effect of informative sampling on sample efficiency by comparing the SRG metric (higher is better) during training for models trained with and without the proposed informative sampling method. The results demonstrate that informative sampling consistently improves the SRG metric across the training process, indicating enhanced sample efficiency.

---

**Algorithm 3** Feature Subset Sampling Using Importance Scores

---

**Require:** $N$: Set of all features, $\phi$: Importance scores for each feature, $n\_masks$: Total number of subsets to sample
**Ensure:** List of sampled subsets
  1: $\phi\_min \leftarrow \min(\phi)$                  ▷ Find minimum feature importance score
  2: $\phi\_max \leftarrow \max(\phi)$                ▷ Find maximum feature importance score
  3: $\phi\_norm \leftarrow \frac{\phi - \phi\_min}{\phi\_max - \phi\_min}$      ▷ Min-max normalization ensuring positive sampling weights
  4: $samples \leftarrow []$                    ▷ List to store sampled subsets
  5: **for** $k = 1$ to $\frac{n\_masks}{2}$ **do**
  6:      $m \leftarrow$ Sample from $U(1, |N|)$      ▷ Sample subset size from uniform distribution
  7:      $S_\phi \leftarrow$ draw $m$ features from $N$ using $\phi\_norm$ as sampling weights, without replacement
  8:      Append $S_\phi$ to $samples$
  9: **end for**
 10: **for** $k = 1$ to $\frac{n\_masks}{2}$ **do**
 11:      $m \leftarrow$ Sample from $U(1, |N|)$      ▷ Sample subset size from uniform distribution
 12:      $S_{1-\phi} \leftarrow$ draw $m$ features from $N$ using $(1 - \phi\_norm)$ as sampling weights, without repl.
 13:      Append $S_{1-\phi}$ to $samples$
 14: **end for**
 15: **return** $samples$

---

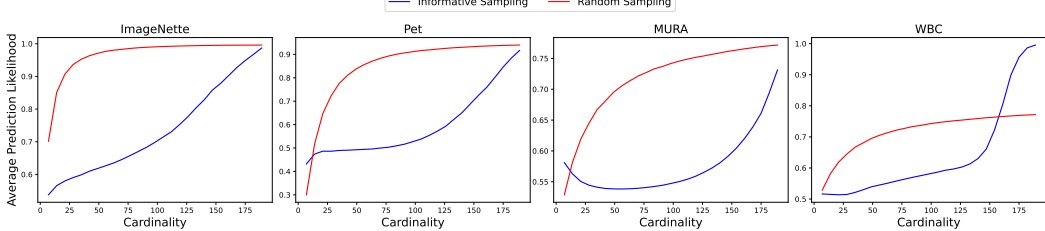

Figure 6: Average prediction likelihood of the ground truth class using informative sampling versus random sampling, computed across various mask sizes.

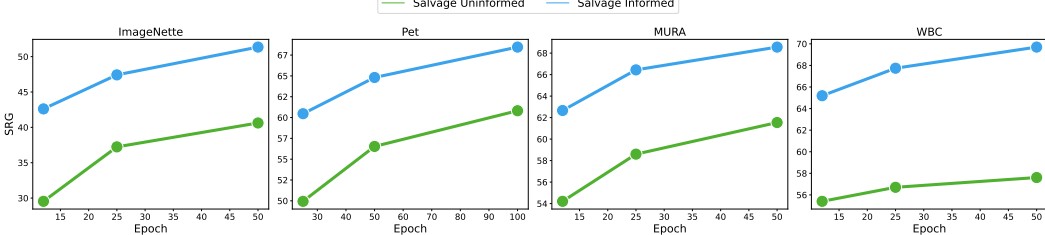

Figure 7: Comparison of the SRG metric during training of the Salvage explainer model. The SRG metric (higher is better) is evaluated for models trained with informative sampling (Salvage Informed) and without it (Salvage Uninformed) across 25%, 50%, and 100% of the overall training duration.

### A.3 FURTHER QUANTITATIVE ANALYSIS

In this section, we present the averaged LIF and MIF scores for all baselines in Table 2, along with the R-LIF and R-MIF scores in Table 3. The results indicate that while the top three methods achieve relatively high LIF and L-LIF scores, our method demonstrates superior performance in the MIF and R-MIF metrics. This suggests that our model is better at identifying a larger portion of the important features, resulting in a clearer distinction between the relevant and irrelevant regions of the image. This observation is in line with the findings of our qualitative results.

Table 2: Quantitative results computed on the dataset Pet, ImageNette, WBC, and MURA. The performance of the 12 compared methods is measured in terms of MIF and LIF.

| | Pet | | ImageNette | | MURA | | WBC | |
| Method | LIF ↑ | MIF ↓ | LIF ↑ | MIF ↓ | LIF ↑ | MIF ↓ | LIF ↑ | MIF ↓ |
|---|---|---|---|---|---|---|---|---|
| GradCam | 81.08 | 70.47 | 88.94 | 90.89 | 78.23 | 62.07 | 76.22 | 94.70 |
| EigenCam | 76.47 | 49.05 | 87.70 | 74.45 | 65.59 | 65.48 | 85.64 | 62.76 |
| Attn. last | 89.44 | 41.55 | 96.76 | 68.74 | 74.69 | 52.28 | 96.46 | 54.27 |
| Attn. Roll. | 89.62 | 37.65 | 96.95 | 64.93 | 73.60 | 56.02 | 97.19 | 49.13 |
| ViT-CX | 88.19 | 37.96 | 96.29 | 66.37 | 74.44 | 54.60 | 93.51 | 51.88 |
| Sal. Maps | 89.38 | 38.26 | 96.09 | 68.34 | 74.57 | 49.31 | 96.36 | 53.58 |
| IntGrad | 89.45 | 62.01 | 96.76 | 85.80 | 75.32 | 61.45 | 97.32 | 85.47 |
| LRP | 89.64 | 40.09 | 97.05 | 69.17 | 75.14 | 55.85 | 97.01 | 60.02 |
| RISE | 95.73 | 32.03 | 98.30 | 75.38 | 93.20 | 36.72 | 98.47 | 77.76 |
| ViT-Shap | 93.52 | 32.45 | 98.71 | 58.38 | 91.25 | 25.91 | 98.81 | 41.38 |
| Salvage | 93.21 | 24.75 | 98.44 | 47.09 | 91.94 | 23.38 | 98.33 | 28.63 |
| Random | 83.29 | 83.67 | 95.13 | 95.07 | 71.24 | 70.78 | 96.55 | 96.55 |

Table 3: Quantitative results computed on the dataset Pet, ImageNette, WBC, and MURA. The performance of the 12 compared methods is measured in terms of R-MIF and R-LIF.

| Method | Pet | | ImageNette | | MURA | | WBC | |
|---|---|---|---|---|---|---|---|---|
| | R-LIF ↑ | R-MIF ↓ | R-LIF ↑ | R-MIF ↓ | R-LIF ↑ | R-MIF ↓ | R-LIF ↑ | R-MIF ↓ |
| GradCam | 85.19 | 81.68 | 92.59 | 95.90 | 78.27 | 68.18 | 77.93 | 98.12 |
| EigenCam | 79.15 | 75.99 | 90.03 | 93.10 | 66.30 | 70.84 | 85.64 | 92.60 |
| Attn. last | 91.40 | 81.78 | 98.47 | 95.44 | 75.36 | 68.36 | 98.95 | 97.34 |
| Attn. Roll. | 90.83 | 79.61 | 98.29 | 94.83 | 74.79 | 68.53 | 99.07 | 96.53 |
| ViT-CX | 89.08 | 71.45 | 97.98 | 90.49 | 74.37 | 65.28 | 98.78 | 91.49 |
| Sal. Maps | 90.84 | 80.06 | 98.09 | 95.25 | 75.19 | 66.69 | 98.94 | 96.83 |
| IntGrad | 90.62 | 82.74 | 98.20 | 96.03 | 75.38 | 69.29 | 99.09 | 97.52 |
| LRP | 91.32 | 82.09 | 98.52 | 95.47 | 75.66 | 68.82 | 99.08 | 97.35 |
| RISE | 91.62 | 73.12 | 98.26 | 92.89 | 81.91 | 59.84 | 98.99 | 95.46 |
| ViT-Shap | 91.66 | 76.97 | 98.89 | 92.67 | 81.05 | 60.47 | 99.55 | 92.28 |
| Salvage | 91.60 | 65.31 | 98.79 | 83.92 | 81.77 | 56.45 | 99.50 | 76.88 |
| Random | 85.20 | 84.91 | 96.66 | 96.68 | 71.53 | 71.79 | 98.15 | 98.15 |

## A.4 CLASSIFICATION ANALYSIS

In this section, we first asses the classification performance of Salvage quantitatively in Table 4 by comparing it to a baseline classifier and the ViT-Shapley explainer model. Salvage demonstrates no major drop in performance relative to the original classifier. In contrast, the ViT-Shapley explainer model performs poorly in classification. This issue can be attributed to the explainer's attribution scores for different classes being decoupled during training due to the use of additive normalization (Covert et al., 2023).

Table 4: An overview of the classification performance of the original classifier, ViT-Shapley, and Salvage computed on Pet, ImageNette, WBC and MURA.

| Model | Pet Accuracy | ImageNette Accuracy | WBC Accuracy | MURA | | | |
|---|---|---|---|---|---|---|---|
| | | | | Precision | Recall | F1-score | MCC |
| Classifier | 95.91% | 99.64% | 99.75% | 84.64% | 78.88% | 81.66% | 0.66 |
| ViT-Shapley | 0.00% | 0.05% | 0.69% | 59.03% | 92.74% | 72.14% | 0.39 |
| Salvage | 93.61% | 98.88% | 99.75% | 80.31% | 80.52% | 80.41% | 0.62 |

## A.5 QUALITATIVE EXAMPLES

We present further qualitative examples comparing Salvage to the baseline methods in Figure 9. The figure includes 5 examples from ImageNette, 5 examples from Pet, and 4 examples from MURA. Additionally we present 8 examples from the WBC Dataset (one example per cell type) in Figure 8.

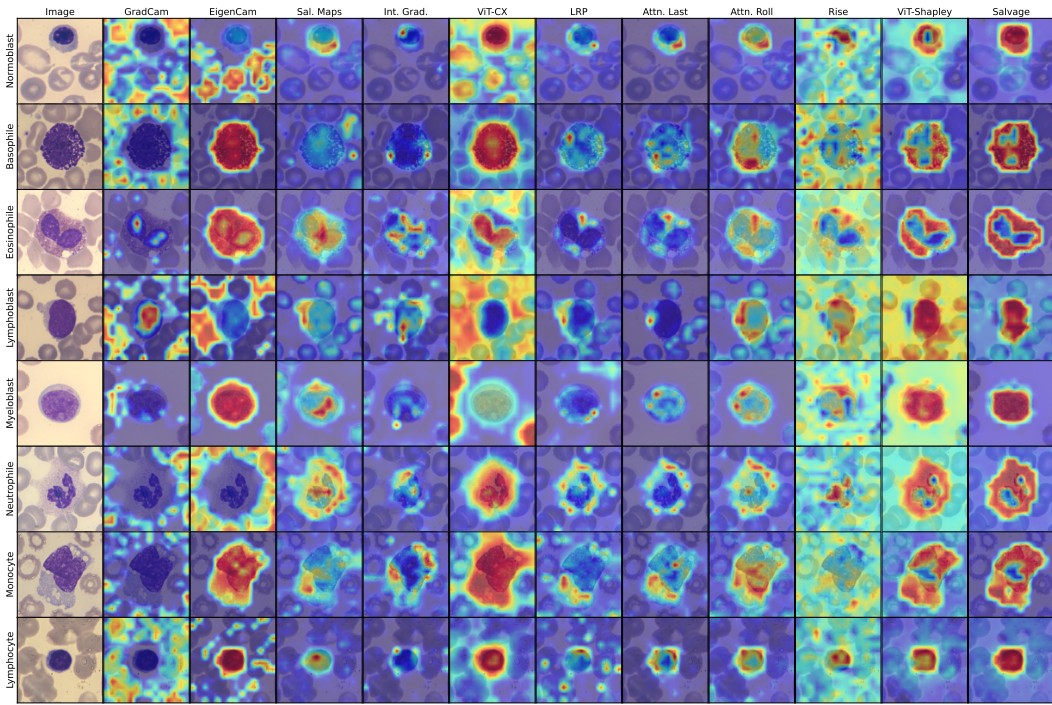

Figure 8: Qualitative examples computed on the WBC dataset.

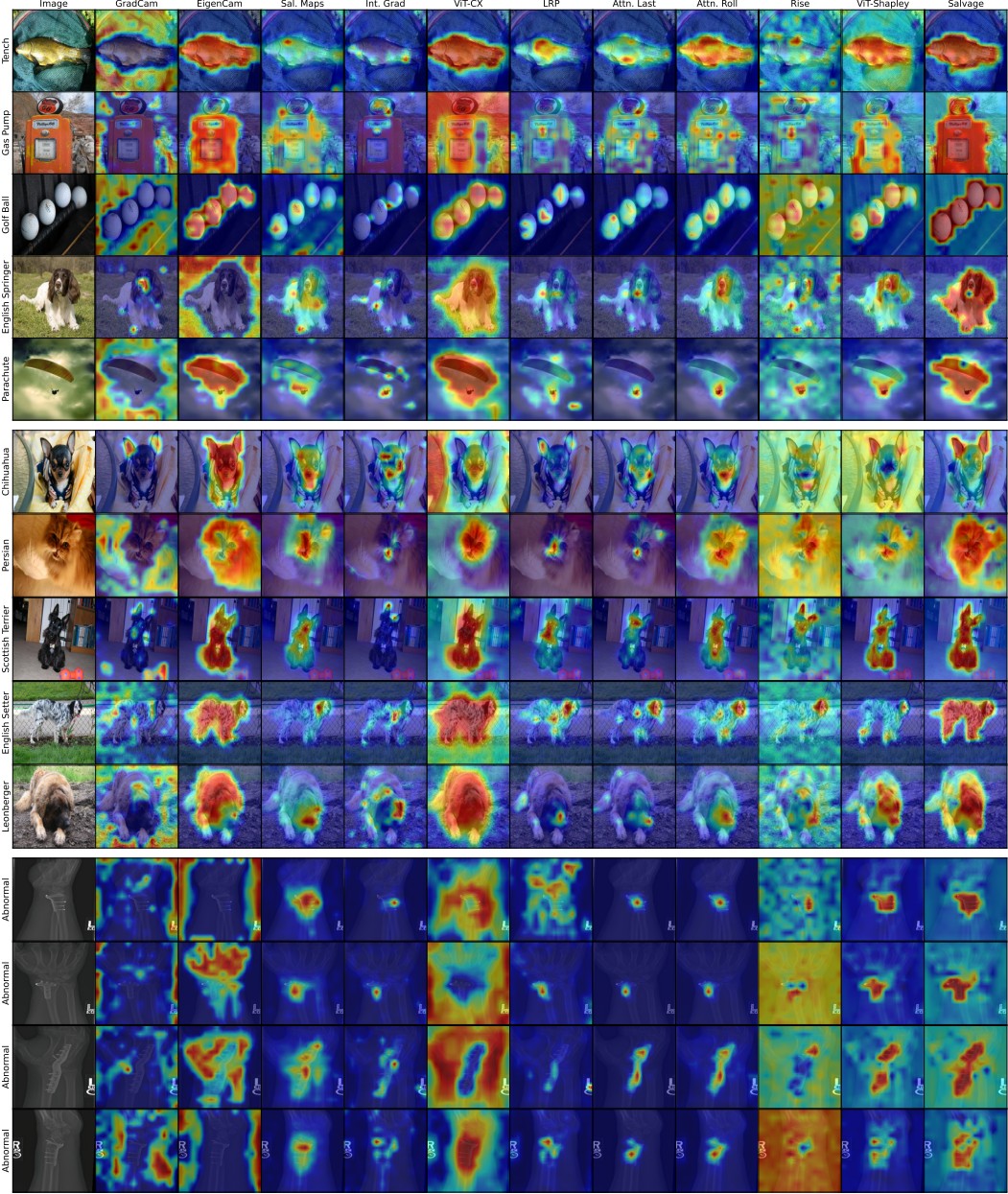

Figure 9: Qualitative examples computed on ImageNette, Pet and MURA.

