# OpenReview forum: "Salvage: Shapley-distribution Approximation Learning Via Attribution Guided Exploration for Explainable Image Classification"
_ICLR.cc/2025/Conference — ICLR 2025 Poster_

### Official Review · Reviewer_Hzto · 2024-10-29

**Soundness:** 4
**Presentation:** 3
**Contribution:** 3
**Rating:** 8
**Confidence:** 3

**Summary:**

This article proposes a new explainability method for image classification. Most of the explainability methods target CNNs; instead, Salvage, the proposed method is architecture agnostic. Salvage is a removal-based approach based on ViT-Shapely with further improvements. The method surpasses the current SOTA.

**Strengths:**

The paper is clear and well-explained. The methodology is adequate. The method is well evaluated. The method, despite not being completely novel, builds on SOTA methods (ViT Shapely) and improves their shortcomings.

**Weaknesses:**

The method is incremental with respect to ViT-Shapely but better. So no much concern.

**Questions:**

I find the paper good as is.

---

> ### Author Response · Authors · 2024-11-22
> **Reply to Reviewer Hzto**
>
> Thank you very much for your positive assessment and for recognizing the strengths of our work. We appreciate your feedback and are glad that the clarity, methodology, and evaluation of the paper met your expectations. Your acknowledgment of our approach to enhancing explainability across architectures encourages us in our efforts to advance robust, architecture-agnostic explainability methods. We have further refined the manuscript based on other reviewer feedback, and we hope these adjustments make our contributions even clearer. A summary of the key revisions will be posted in a general comment. We mark changes in blue in the manuscript. Thank you again for your support and recommendation.

---

> > ### Comment · Reviewer_Hzto · 2024-11-26
> > **Response**
> >
> > Thanks to the reviewers and authors for their feedback. I agree with the points brought by other reviewers, like missing ablation studies, other comparisons, incomplete results, or using other base models. However, after reading the authors' responses to these concerns, I find that they have given an adequate answer to all of these concerns. So, if the other reviewers agree that they have answered correctly to their concerns, I would be happy to maintain my score.

---

> ### Author Response · Authors · 2024-11-26
> **Reply to Reviewer Hzto (2)**
>
> Thank you again for your positive feedback and encouraging assessment of our work. We wanted to inform you that we have just uploaded an improved version of the paper, further refining its content and presentation.
>
> In this revision, we have focused on enhancing the clarity and presentation of the classification analysis of our method. Additionally, we have included an evaluation on a new dataset, the WBC Dataset, to further emphasize the robustness of our method and its applicability to diverse image types.
>
> To help identify changes, all modifications made in this second revision are marked in dark green, while changes from the first revision remain in dark blue. We hope these updates further strengthen the presentation and contributions of our work.

---

### Official Review · Reviewer_7jFJ · 2024-11-02

**Soundness:** 2
**Presentation:** 2
**Contribution:** 2
**Rating:** 6
**Confidence:** 3

**Summary:**

This paper introduces Salvage,a removal-based explainability method for image classification. It includes a concept of Shapley-distributions,which offers a more accurate approximation of classification probability distributions and an informed sampling strategy that leverages approximated feature importance scores to reduce imbalance and facilitate the estimation of underrepresented features.

**Strengths:**

1. A  new concept of Shapley-distributions,which offers a
 more accurate approximation of classification probability distributions, is introduced.
2. The comparison methods, datasets, and metrics are quite comprehensive. On some metrics, it has a clear advantage.

**Weaknesses:**

**The presentation should be improved a lot.**

1. The explanation of symbols in the formulas is not clear enough, causing difficulty in understanding, such as Eq.2.
2. In Table 1, there are several obvious typographical errors of the experimental results, for example “68,56”. It should be “68.56”.
3. The bottom line of the Table 2 is not drawn!

**Soundness**
1. There is no theoretical  proof or experimental results can demonstrate that the INFORMATIVE SAMPLING has improved efficiency.
2. In Table 2, on PET, in terms of RRA, Salvage underperforms about 3 point compared with SOTA. There should be appropriate analysis and discussion regarding this.
3. Only make ablation study on INFORMATIVE SAMPLING.

**Questions:**

1.How is the experimental performance when using SHAPLEY DISTRIBUTION ESTIMATION alone?

2. Is this method applicable to other types of tasks (such as object detection or segmentation)?

---

> ### Author Response · Authors · 2024-11-22
> **Reply to Reviewer 7jFJ**
>
> We thank the reviewer for the time and effort spent to review our manuscript. We believe the constructive feedback led to an improvement of the manuscript, and we will discuss each raised point in turn. We mark changes in blue in the manuscript.
>
> **Weaknesses:**
>
> **Presentation**
>
> 1. Thank you for pointing out the need for clearer explanations of "p_w(S) ∝ w_S"  in Eq. 2. In response to your feedback, we have revised the manuscript to explicitly define “p_w(S) ∝ w_S"​, clarifying that this notation denotes a feature subset distribution sampled proportionally to w_S (line 164-165). We hope this change enhances the clarity of our presentation and makes the methodology easier to understand.
> 2. Thank you for noticing these typos. We have corrected them in Table 1 to ensure consistency in our presentation of results.
> 3. Thank you for pointing this out. We have added the missing bottom line to Table 2 to improve its completeness and visual clarity.
>
> **Soundness:**
>
> 1. Thank you for pointing out the missing experimental results supporting the claim about efficiency improvement through informative sampling. We included an additional study in section A.2 (line 749-754).  In particular, we illustrate in Figure 6 the effect of informative sampling on sample efficiency by comparing the explanation performance of Salvage during training for models trained with and without the proposed informative sampling method. The results demonstrate a faster convergence of our explainer model to a better performance when using informative sampling, highlighting its enhanced sample efficiency.
> 2. Thank you for highlighting this aspect. The methods with the highest RRA scores indeed achieve a slightly stronger alignment in ranking, meaning their top attribution scores lie within the object of interest. However, these methods also show a lower attribution mass within the object (RMA) and exhibit weaker SRG and R-SRG scores, suggesting they may not focus on the most relevant parts of the object for the prediction, potentially limiting their effectiveness.
> In contrast, Salvage achieves a balanced performance across all metrics, with a higher RMA score that indicates a more concentrated attribution mass within the object of interest, as well as stronger SRG and R-SRG scores. This combination highlights Salvage’s effectiveness in identifying not only the presence of the object but also the most relevant regions within it, which is essential for meaningful explainability. We believe this multi-metric approach offers a more comprehensive perspective when evaluating explainability methods. We have updated our results in the discussion in the revised manuscript to reflect this comprehensive, multi-metric perspective (line 388-392).
> 3. Thank you for your comment regarding the ablation study on informative sampling. In order to address your issue and enhance our analysis, we revised our ablation study section (Section 5.4: line 421-431) and Figure 3, separating the ablation studies of the informative sampling and Shapley distribution estimation.  This structure provides a clearer view of the contributions from each component individually. We hope this update sufficiently addresses your concerns and clarifies our analysis.
>
> **Questions:**
> 1. Thank you for this question. We have included these results in the revised ablation study (Section 5.4: line 421-431). In particular, we assess the performance gain of using Shapley distribution estimation compared to an MSE-based approximation (ViT-Shapley) without informative sampling in Figure 3. The results show that Shapley distribution estimation clearly outperforms MSE-based approximation (ViT-Shapley) on ImageNette and Pet datasets, but shows a slightly lower performance on MURA.  However, we believe the latter could benefit from adjusting the temperature parameter in the sigmoid function used during training.
> 2. The concept of Shapley distribution can be applied to any classification setting:
> For semantic segmentation, which involves pixel-level classification, our method’s principles can be applied to estimate attribution scores for each pixel. However, estimating these scores in a segmentation setting is computationally intensive, as it requires calculating an attribution map for each single pixel-class pair. This makes semantic segmentation a challenging domain for explainable AI, which is why it still remains largely unexplored. Applying Salvage to segmentation would require an efficient architecture capable of handling the substantial quadratic increase in computation.
> For object detection, additional challenges arise due to the diversity of the used frameworks and architectures. An extension of Salvage could potentially be developed to explain the classification of detected objects, though modifications would be required to handle the distinct structure of object detection models.

---

> > ### Comment · Reviewer_7jFJ · 2024-11-26
> >
> > Most issues have been resolved, and I have accordingly increased the score.

---

> > > ### Author Response · Authors · 2024-11-26
> > > **Reply to Reviewer 7jFJ (2)**
> > >
> > > We thank you again for your valuable feedback to improve our paper and fair assessment. We uploaded an improved version of our paper mainly focusing on the presentation of the classification analysis of our method and including a new dataset in our analysis to further highlight the robustness of our method.  All changes in the manuscripts in our second revision are in dark green, while we kept the changes from the first revision in dark blue.

---

### Official Review · Reviewer_sz6D · 2024-11-03

**Soundness:** 2
**Presentation:** 3
**Contribution:** 3
**Rating:** 6
**Confidence:** 3

**Summary:**

This paper introduces a novel explainability method for image classification known as Salvage. The paper employs a removal-based technique coupled with the concept of Shapley-distributions. These techniques are used to train an explainer model that learns the prediction distribution of the classifier on masked images. The authors address the imbalance between important and unimportant features by devising an informed sampling strategy. This strategy facilitates better approximation of the classifier’s distribution and helps the estimation of underrepresented features. The effectiveness of Salvage is validated through experiments on the ImageNette, MURA, and Pet datasets. The study illustrates that Salvage outdoes various baseline explainability methods and can additionally be used as a fully explainable classifier without a considerable fall in classification performance. The paper concludes by pointing out future optimizations and improvement possibilities for Salvage.

**Strengths:**

The paper presents "Salvage," a novel removal-based explainability method for image classification that tackles unbalanced important features via an informed sampling strategy. The invention of Shapley-distributions for a more accurate approximation of classification probability distributions is impressive. The paper's comprehensive and clear presentation, alongside robust experimental evaluation, underlines the method's potential for explainable classification, with comparable accuracy to standard classifiers.

**Weaknesses:**

1. Although the paper demonstrates a performance comparison with a couple of explainability methods like ViT-Shapley and RISE, more extensive comparison with a wider array of contemporary removal-based explainability methods could provide a more robust evaluation of the Salvage algorithm's efficacy.
2. It would be beneficial to see Salvage's performance with other types of data or in other domains, beyond the ones mentioned in the paper (ImageNette, MURA, and Pet datasets). This would help in evaluating the broad applicability and versatility of the approach.
3. The impacts of temperature parameter changes in the softmax or sigmoid functions during the approximation of the classifier’s distribution could have been explored in more depth. More extensive experimental study in this aspect could enhance the robustness of Salvage.

**Questions:**

1. Could you please elaborate more on why the informative sampling's improvement in performance is less noticeable in the Pet dataset compared to the other datasets?
2. How would different neural architecture designs for the explainer model impact the performance of Salvage?
3. Given that the model has been tested on a limited number of datasets, have you considered testing Salvage on a wider array of datasets, particularly more complex or diverse ones, to evaluate its broad applicability?

---

> ### Author Response · Authors · 2024-11-22
> **Reply to Reviewer sz6D**
>
> We would like to begin by expressing our sincere gratitude to the reviewers for their thoughtful and constructive feedback. We truly appreciate the time and effort taken to evaluate our work, and we are grateful for the valuable insights that have helped us strengthen our paper. We mark changes in blue in the manuscript.
>
> **Weaknesses:**
>
> 1. Thank you for pointing this out. In our comparison, we included 10 diverse explainability baselines covering gradient-, attention-, LRP-, and removal-based methods. ViT-Shapley and RISE were selected as the removal-based baselines because they represent state-of-the-art removal-based approaches for image classification attribution, known for their robustness and widespread use as benchmarks. Many other removal-based methods, while valuable, are not scalable to settings with a large number of features, such as image classification tasks, where efficiency becomes a critical factor. We believe that ViT-Shapley and RISE provide a comprehensive and balanced overview of the SOTA removal-based methods, without introducing redundancy.
>
> 2. See Question 3.
>
> 3. Thank you for this insightful suggestion. We agree that adjusting the temperature parameters in the softmax and sigmoid functions could further enhance Salvage's performance, as suggested in our future work section (line 532-535). While systematic tuning of these parameters has potential benefits, it also introduces significant computational overhead. In this study, our primary goal was to deliver an effective attribution method for image classification within a reasonable computational budget, avoiding additional hyperparameter optimization. This approach allowed us to achieve state-of-the-art performance while ensuring fairness and applicability by staying within the computational constraints of existing methods.
>
> **Questions:**
> 1. Thank you for this thoughtful question. We hypothesize that this may be attributed to the ratio between important and unimportant features (image patches) within the images; since the imbalance between informative and uninformative features can vary significantly across datasets, the impact of informative sampling is more pronounced in datasets where this imbalance is greater. Consequently, a smaller imbalance between the number of informative and uninformative features within the Pet dataset could be the subject of a less noticeable performance gain through informative sampling.
> 2. Thank you for this insightful question. The explainer model in Salvage was designed to estimate attribution scores for each superpixel-class pair, a task very similar to image segmentation. Therefore, we anticipate that improved segmentation architectures would likely enhance performance in accurately learning attribution scores, as they are generally better suited to capture spatial relationships effectively. We appreciate the opportunity to consider this direction in future work and include its motivation in the manuscript (Section 7: line 530-532).
> 3. We appreciate the reviewer's suggestion to evaluate Salvage on a broader range of datasets, as a diverse and extensive evaluation is crucial for demonstrating its wider applicability. In this study, we aimed to provide a robust evaluation by comparing  Salvage to 10 baselines from different methodologies using 6–8 distinct metrics across three datasets. These datasets were carefully selected to represent diversity in both the number of classes (2, 11, and 37) and the domain (real-world objects, pets, and radiology images), providing a robust evaluation of the model's performance in various settings.
> While we recognize that adding more datasets could further enhance the comprehensiveness of our analysis, the computational demands of reporting multiple metrics for each method across additional datasets are substantial. To increase the diversity of our analysis, we will include one additional benchmark and will add these results to the appendix shortly.

---

> > ### Author Response · Authors · 2024-11-26
> > **Reply to Reviewer sz6D (2)**
> >
> > We thank the reviewer again for the valuable and thoughtful feedback. We have uploaded a new revision to address your concerns about the dataset diversity in our study
> >
> > **Question 3:**
> >
> > To address your suggestion to increase dataset diversity, we have incorporated an evaluation on the WBC Dataset [1], which focuses on classifying various types of pathological and normal white blood cells. We believe this dataset significantly enhances the diversity of our evaluations, as it introduces a new type of image data that is markedly different from the other three datasets. This addition provides a broader perspective on the applicability of our method.
> > To help distinguish changes across revisions, modifications introduced in our second revision are highlighted in dark green, while changes from the first revision remain in dark blue.
> >
> >
> > [1] Alexandra Bodzas, Pavel Kodytek & Jan Zidek. A high-resolution large-scale dataset of pathological and normal white blood cells. Scientific Data, 10, 07 2023. doi: 10.1038/s41597-023-02378-7.

---

> > > ### Comment · Reviewer_sz6D · 2024-11-28
> > >
> > > Thank you for the authors' detailed response. Considering the presentation and overall contribution of the paper, I will keep my score at 6.

---

### Official Review · Reviewer_d1Q4 · 2024-11-04

**Soundness:** 3
**Presentation:** 3
**Contribution:** 3
**Rating:** 6
**Confidence:** 4

**Summary:**

The authors highlight the shortcomings of existing methods for existing Shapley-based explainers and propose a method called Salvage, which effectively learns and samples based on the Shapley value distribution.

**Strengths:**

1) The issues identified with existing methods appear valid and relevant.
2) The authors demonstrate an improvement in explanation accuracy compared to existing methods across various datasets.

**Weaknesses:**

1) The explainer model is essentially an estimation model for interpreting the behavior of the target classification model; however, this paper does not clearly define what the target model is. Furthermore, there is insufficient evidence to show that the method operates effectively across different target models.
2) Some results seem incomplete, as suggested by Figure 1.
3) The ablation study is lacking. While the proposed method focuses on effectively learning the distribution and sampling, there is no analysis of which aspect is more critical to the overall success of the method.
4) The problems identified with existing methods are described conceptually but lack empirical validation.

**Questions:**

1) What exactly does Figure 3 illustrate?
2) Since the goal is to derive an explainer model for a specific classification model, I am curious not only about the classification performance but also about how well the predictions align with those of the existing classification model (as seen in Table 2).

---

> ### Author Response · Authors · 2024-11-22
> **Reply to Reviewer d1Q4**
>
> We first want to extend our sincere thanks to the reviewer for their excellent and constructive feedback. We greatly appreciate the time and effort invested in evaluating our work. We mark changes in blue in the manuscript.
>
> **Weaknesses:**
> 1. Thank you for your thoughtful feedback. We conducted analyses using three distinct targets: specifically one respective Vision Transformer classifier for each of the three datasets – ImageNette and Pet datasets for multi-class classification and the MURA dataset for binary classification. We further clarified this in the manuscript in Section 5.1 (line 265-267). Additionally, we are currently running new experiments to further demonstrate the effectiveness of our method across an extended set of image domains, to provide even broader evidence of our method’s applicability. Does this address your concerns regarding model diversity?
>
> 2. Thank you for highlighting this issue. In the initial submission, we included 7 out of our 10 baselines in the qualitative examples for enhanced readability, but we understand this may have given the impression of an incomplete analysis. We have now updated our manuscript to include all methods in the qualitative examples of the main paper in Figure 1 and the appendix (A.5) in Figure 8. Additionally, for quantitative results, the full set of the missing metric scores (LIF, MIF, R-LIF, and R-MIF) was included in the appendix (A.3) in Tables 3 and 4 due to space limitations, ensuring the completeness of our analysis while maintaining clarity in the main text.
>
> 3. See Question 1.
> 4. Thank you for this valuable feedback. In order to support our claims about problems identified with existing methods, we revised Section 5.4 (lines 421-450) and Figure 3 and included further investigations of informative sampling in Section A.2 (line 749-753). In our paper, we aimed to address two primary issues:
>    *  Limitations of MSE-based probability distribution approximations: In order to demonstrate these limitations, we conducted an ablation study comparing the baseline using MSE-based approximation (ViT Shapley), to our alternative solution (Shapley distribution estimation) without informative sampling. The ablation results are shown in Figure 3 and discussed in Section 5.4 (line 427-430), demonstrating a major performance improvement when using Shapley distribution estimation on Pet and ImageNette.
>
>    * Imbalance of prediction likelihood with uniform sampling and its impact on performance: In Section A.2 (line 743-754), we first illustrated how prediction likelihoods become more balanced when using informative sampling in Figure 5. We then show how addressing this unbalance enhances the sample efficiency throughout training by comparing models trained with informative sampling (Salvage Informed) and without it (Salvage Uninformed) across 25%, 50%, and 100% of the overall training duration in Figure 6. Moreover, we conducted an ablation study comparing the final performance of our model with and without informative sampling in Section 5.4.
>
> **Questions:**
> 1. Thank you for raising this point. We absolutely agree on the importance of quantifying the relative importance of distribution learning versus informative sampling in our method’s success. To address this, we revised Section 5.4 (line 420-431) and Figure 3 to investigate the individual contribution of each core principle in our method. The first study evaluates the performance gain of using Shapley distribution estimation compared to an MSE-based approximation (ViT-Shapley) without informative sampling. The second study examines the impact of informative sampling by comparing Salvage’s performance with and without it. The results, illustrated in Figure 3, indicate that the success of our method can not be addressed to a single one of the presented method’s principals. For example, Shapley distribution estimation significantly improved performance on the Pet dataset, whereas Informative Sampling had little impact; conversely, Informative Sampling led to notable gains on MURA, whereas Shapley distribution estimation did not.
> 2. Thank you for this insightful suggestion! We have gladly included this analysis in the appendix (new Section A.4: line 864-890) to provide more insights about the alignment between the predictions of the explainer model and those of the original classification model. In Figure 7, we present the confusion matrices for the explainer and classifier, focusing on the overlap between correctly and wrongly classified samples.

---

> > ### Comment · Reviewer_d1Q4 · 2024-11-25
> >
> > Thank you for the detailed response to my initial review. Most of my concerns have been addressed by the authors’ response; however, there are still a few points that remain unclear.
> >
> > **Weakness 1**
> >
> > The authors mention experimenting with ViT, but they do not specify which version of ViT (or detailed parameters) was used. It is important to clarify this detail. Additionally, ensuring diversity not only in the datasets but also in the models tested is crucial for the robustness of the results.
> >
> > **Question 2**
> >
> > Thank you for revising the paper based on my feedback. One additional comment: Since the goal is to derive an explainer model for a specific classification model, I believe that the alignment between the explainer model and the classification model is more important than the absolute classification performance. In other words, the importance of Figure 7 outweighs that of Table 2. It would be more effective to compare ViT-Shapley with the results shown in Figure 7, similar to the approach in Table 2, and to move this comparison to the main body of the paper.
> >
> > I have increased the score accordingly.

---

> > > ### Author Response · Authors · 2024-11-26
> > > **Reply to Reviewer d1Q4 (2)**
> > >
> > > Thank you for your prompt and constructive feedback to improve our paper.
> > >
> > > **Weakness 1:**
> > >
> > > Thank you very much for further clarifying your concern. We apologize for initially misunderstanding your point. In response, we have updated the experimental section to include the missing details, specifying that we use a Vision Transformer base model (ViT-B) with a patch size of 16 and registers, leveraging the implementation and pre-trained weights provided by DINOv2 (lines 268–271).
> > > We completely agree that additional analysis across different target models could further validate the robustness of our method. Unfortunately, we are unable to include such an analysis within the remaining discussion period. However, as our method doesn’t make any assumption about the architecture and size of the target model, we expect that its performance shouldn’t be affected by the choice of the target model.
> > >
> > > **Question 2:**
> > >
> > > Thank you for your thoughtful comment! We would like to emphasize the reasoning behind highlighting the accuracy of our explainer model when used as a classifier. Using the explainer as a unified model for both classification and explanation ensures that classification and explanations remain consistent, even under domain shift. This highlights the potential of our method for explainable classification without the need for the initial classifier model in test-time.
> > > Conversely, when using the classifier for classification and the explainer for explanation, the overlap between their predictions becomes an important aspect. It serves as an indicator of the alignment between the decision-making processes of the classifier and the explainer.
> > > Regarding your suggestion, we agree and have moved the confusion matrix figure (now Figure 4) to the main paper while relocating the classification performance Table (now Table 4) to the appendix. The confusion matrix provides valuable insights into both the classification performance of Salvage and its alignment with the classifier model's predictions. We have also updated the heatmap to average the cell values over the total number of predictions, reflecting both accuracy and prediction overlap.
> > >
> > > **Information regarding the new revision:**
> > >
> > > We uploaded a new revision to address your concerns as mentioned. We additionally included a fourth dataset to our analysis and updated all tables and figures with the results. All changes in the manuscripts in our second revision are in dark green, while we kept the changes from the first revision in dark blue.
> > >
> > > We thank you again for your constructive feedback and fair assessment.

---

> ### Author Response · Authors · 2024-12-03
> **Reply to Reviewer d1Q4 (3)**
>
> Thank you again for your valuable feedback, which has strengthened our work.
> We appreciate the extension of the discussion phase, which allowed us to address your final concern regarding the models diversity through additional experiments:
>
> **Weakness 1:**
>
> To demonstrate the robustness of our method across different architectures, we evaluated its performance on the convolutional neural network (CNN) architecture EfficientNetV2. Given that our initial benchmark included a transformer-based architecture, we selected a CNN to maximize diversity between the evaluated models.
>
> We report the results for our method and seven baselines (Grad-CAM, Eigen-CAM, Saliency Map, Integrated Gradients, RISE, and ViT-Shapley) on the following datasets: Pet, ImageNette, and WBC. Note that ViT-CX, Attention Rollout, LRP (beyond attn), and Last-Attention were excluded from this evaluation as they are not applicable to CNN architectures.
>
> Although it is not possible to modify the manuscript at this stage, these results will be included as an appendix section in the camera-ready version upon acceptance.
>
> | Method       | Pet (SRG) | Pet (R-SRG) |Pet (RMA) | Pet (RRA) | WBC (SRG) | WBC (R-SRG) | Imagenette (SRG) | Imagenette (R-SRG) |
> |--------------|-----------|-------------|----------|-----------|-----------|-------------|------------------|--------------------|
> | GradCam      | 38.89     | 11.83       | **60.11**    | 69.53     | 32.45     | 3.47        | 17.08            | 4.11               |
> | EigenCam     | 30.12     | 24.14       | 45.21    | 57.27     | 17.20     | 17.47       | 13.45            | 8.73               |
> | VanillaGrad  | 28.47     | 5.75        | 41.15    | 62.67     | 9.79      | 0.01        | 8.87             | 1.21               |
> | IntGrad      | -2.26     | 0.50        | 43.75    | 39.01     | -1.58     | -0.60       | 2.00             | -0.14              |
> | Rise         | 51.73     | 16.01       | 30.10    | 41.75     | 16.56     | 2.53        | 21.05            | 5.31               |
> | ViT-Shapley  | 26.10     | 5.24        | 32.34    | 57.69     | 51.09     | 5.28        | 30.84            | 5.09               |
> | Salvage      | **58.99**     | **24.33**       | 59.00    | **77.64**     | **63.11**     | **19.49**       | **41.40**            | **10.67**              |
>
> The results demonstrate that our method outperforms all baselines, also when applied to a CNN architecture, further underscoring its robustness across diverse model architectures.

---

### Author Response · Authors · 2024-11-22
**General Reply (1)**

First and foremost, we thank all the reviewers for their time and effort in reviewing our manuscript. The constructive feedback has significantly contributed to improving our paper. We have submitted a revised version with all changes marked in blue. Below each review, we provide detailed responses to the raised points. The key changes are as follows:

* We added qualitative examples of the missing baselines to Figures 1 and 8.
* We revised Section 5.4 and Figure 3 to provide clearer ablation studies investigating the individual contribution of each core principle in our method (Shapley distribution estimation and informative sampling).
* An efficiency analysis of Salvage with and without informative sampling has been added to Section A.2 of the appendix.
* We added a new Section A.4 to the appendix providing more insights about the alignment between the predictions of the explainer model and those of the original classification model.
* We improved upon the presentation of the paper.

We hope these revisions address the concerns raised, and we thank the reviewers again for the constructive and fruitful discussion.

---

### Author Response · Authors · 2024-11-26
**General Reply (2)**

We would like to thank all reviewers once again for their valuable feedback and fair assessment. In response, we have submitted a revised version of the manuscript, with all new changes highlighted in green while retaining changes from the first revision in blue. The key updates are as follows:

- **Inclusion of an additional dataset (WBC):** We incorporated the WBC dataset [1]. We believe this dataset significantly enhances the diversity of our evaluations, as it introduces a new type of image data that is markedly different from the other three datasets.  We updated all relevant tables and figures to include its results. This addition aims to further demonstrate the robustness of our method across diverse datasets, addressing a suggestion from reviewer sz6D.

- **Improved presentation of the classification analysis (Section 5.5):** We enhanced the discussion to include both the classification accuracy of Salvage and the alignment of its predictions with the classifier model. This refinement follows reviewer d1Q404's suggestion to provide a more comprehensive analysis.
We hope these updates address the reviewers' suggestions and further clarify the contributions of our work. Thank you again for your constructive feedback and thoughtful evaluation.

[1] Alexandra Bodzas, Pavel Kodytek & Jan Zidek. A high-resolution large-scale dataset of pathological and normal white blood cells. Scientific Data, 10, 07 2023. doi: 10.1038/s41597-023-02378-7.

---

### Meta-Review · Area_Chair_tmG1 · 2024-12-20

**Metareview:**

This paper proposes Salvage, a removal-based explainability method for image classification. The proposed method uses the concept of Shapley-distribution by learning the prediction distribution on masked images. This strategy helps the estimation of underrepresented features. Experiments were conducted on ImageNette, MURA and Pet datasets.

Overall, after the author-reviewer discussion, all the reviewers agreed that this paper exceeds the acceptance threshold. Specifically, the authors' responses include more experimental results on an additional dataset (WBC) and the revised paper's presentation is improved based on the reviewers' comments. I agree with the reviewers and recommend acceptance.

**Additional Comments On Reviewer Discussion:**

- [d1Q4, 7jFJ] There were a few concerns regarding presentation
- [d1Q4, 7jFJ] The reviewers pointed out that the ablation study is lacking
- [d1Q4, sz6D] Not enough comparison with the existing methods
- [sz6D] Experiments on other datasets and other domains are required
- [7jFJ] There is not enough justification for how informative sampling improves efficiency

The authors addressed the concerns by revising the paper with a better presentation and adding additional experiments, e.g, ablation studies and WBC experiments.

---

### Decision · Program_Chairs · 2025-01-22

Accept (Poster)